# The burden of generational harm due to alcohol use in Tanzania: A mixed method study of pregnant women

**Kirstin West**[1], **Alena Pauley**[1], **Mia Buono**[1], **Miriana Mikindo**[2], **Yvonne Sawe**[2], **Joseph Kilasara**[2,3], **Francis Sakita**[4,5], **Sharla Rent**[1,6], **Bariki Mchome**[5,7], **Blandina T. Mmbaga**[1,2,4], **Catherine A. Staton**[1,8]*

**1** Duke Global Health Institute, Duke University, Durham, North Carolina, United States of America, **2** Kilimanjaro Clinical Research Institute, Moshi, Tanzania, **3** Department of Clinical Nursing, Kilimanjaro Christian Medical University College, Moshi, Tanzania, **4** Kilimanjaro Christian Medical Center, Moshi, Tanzania, **5** Kilimanjaro Christian Medical University College, Moshi, Tanzania, **6** Duke Department of Pediatrics, Duke University School of Medicine, Durham, North Carolina, United States of America, **7** Department of Obstetrics and Gynecology, Kilimanjaro Christian Medical Center, Moshi, Tanzania, **8** Department of Emergency Medicine, Duke University School of Medicine, Durham, North Carolina, United States of America

* catherine.staton@duke.edu

## Abstract

### Background

Rates of prenatal alcohol use in Sub-Saharan Africa (SSA) are increasing despite regulatory bodies urging pregnant women to abstain from alcohol. Tanzania has minimal policies, interventions, and educational programs addressing prenatal alcohol exposure. Consequently, a considerable number of mothers and their fetuses are exposed to alcohol, leading to serious health consequences like fetal alcohol spectrum disorder (FASD). Our study aims to understand the alcohol use practices of pregnant women, the knowledge and attitudes related to prenatal alcohol exposure among different genders and generations, and how these may be influenced by community perceptions and cultural beliefs among patients at Kilimanjaro Christian Medical Center (KCMC).

### Methods

A total of 676 individuals seeking care at the KCMC Emergency Department (ED) or Reproductive Health Center (RHC) met our inclusion criteria, were approached for participation using a systematic random sampling method, and were enrolled. Of those, 541 women and 114 men completed the survey. The quantitative analysis focused exclusively on survey data from 533 women who reported their pregnancy status and age. Descriptive frequencies were used to compare sociodemographic factors and alcohol use practices across three female groups stratified by pregnancy status and age. Nineteen survey participants—both men and women—were purposively selected for qualitative semi-structured in-depth interviews (IDIs) exploring knowledge, attitudes, and cultural beliefs surrounding alcohol use during pregnancy. A grounded theory approach was used to analyze IDIs in NVivo.

**Data availability statement:** Data are only available upon reasonable request, as participants did not consent to public data transfer and requires a written agreement approved by Kilimanjaro Christian Medical Centre Ethics Committee and the National Institute for Medical Research (Tanzania). Data inquiries can be sent to Gwamaka W. Nelson at gwamakawilliam14@gmail.com.

**Funding:** This project was funded by the Duke Global Health Institute Graduate Student funds (AMP), and the Josiah Trent Foundation (21-06 to CAS). These two financial awards funded the salaries of JK, YS, and MMi as research assistants hired specifically for this study. No other authors received specific funding for this work. Infrastructure built by NIH grants (R01 AA027512 to CAS) was used to support the data collection and analysis processed for this grant to understand gender-related aspects of alcohol use at KCMC. The funders had no role in study design, data collection and analysis, decision to publish, or preparation of the manuscript.

**Competing interests:** The authors have declared that no competing interests exist.

## Results

A large percentage of pregnant women reported consuming alcohol at least once per week (42.2%). Older non-pregnant women exhibited the highest rate of alcohol use per week (66.0%). Many older non-pregnant women (28.7%) perceived alcohol use as acceptable during pregnancy. Younger non-pregnant women had the highest prevalence of harmful or hazardous drinking (HHD) at 16.4%. This group also reported the highest weekly alcohol expenses, with 18.1% spending over 10,000 Tanzanian shillings (TZS) per week. Median [IQR] DrInC scores were 0 [0–0] for pregnant women, 0 [0–7] for younger non-pregnant women, and 0 [0–1] for older non-pregnant women. Older non-pregnant women exhibited the highest prevalence of depression (31.4%). Median [IQR] PHQ-9 scores were 4 [3–6.25] for pregnant women, 5 [2–8] for younger non-pregnant women, and 6 [3–10] for older non-pregnant women. Qualitative analyses demonstrated that (1) cultural beliefs are intricately tied to perceived benefits of prenatal alcohol exposure, (2) a history of alcohol use preceding pregnancy largely influences prenatal alcohol use, and (3) community views of PWCA are negative.

## Conclusions

Our findings demonstrate high rates of prenatal alcohol use in Moshi, Tanzania, with pre-pregnancy alcohol use as a significant predictor. Despite generally negative views of pregnant women who consume alcohol (PWCA), some sociocultural beliefs and limited knowledge about the dangers of prenatal alcohol exposure encouraged alcohol use during pregnancy. To improve health outcomes and reduce alcohol-related pregnancy complications for current and future generations, community-wide health messaging and pre-pregnancy interventions may prove beneficial for pregnant women and women of reproductive age who consume alcohol.

## 1. Introduction

As alcohol use increases globally, so do the number of alcohol-related health consequences, including death and disability [1,2]. In a 2020 global risk assessment, alcohol consumption ranked among the top ten factors impacting overall health and served as a major risk factor for communicable, maternal, perinatal, and nutritional diseases [2].

A notable, well-established alcohol-related harm is the detrimental effect on pregnancy [2–4]. Alcohol use during this period can cause an increased risk of congenital and developmental disabilities, miscarriage, intrauterine growth restriction, spontaneous abortion, preterm labor, and stillbirth [3,4]. Global estimates from 1984 to 2014 indicated that approximately 9.8% of pregnant women worldwide consumed alcohol [5]. Particularly prevalent in low- and middle-income countries, rates of alcohol consumption during pregnancy can range from 2.5% in Nigeria to as high as 32.5% in the Democratic Republic of the Congo [6]. Fetal alcohol spectrum disorder (FASD) encompasses a range of health issues that children may develop due to intrauterine alcohol exposure, including growth retardation, facial dysmorphology, cognitive dysfunction, and neurobehavioral disabilities [7]. FASD is often under diagnosed in Africa, with prevalence rates in South Africa ranging from 16% to 31% [8,9]. The WHO African region was found to have the highest prevalence of binge drinking, defined as having four or more drinks on a single occasion, during pregnancy, estimated at 3.1% [10]. Populations with high rates of binge drinking have been linked to similarly high rates of FASD [11].

Existing literature emphasizes the profound health consequences of prenatal alcohol exposure globally and in sub-Saharan Africa (SSA) [3,5,6,10–13], necessitating increased awareness and robust intervention strategies [14–16]. The World Health Organization (WHO) urges pregnant women to abstain from alcohol use due to the health risks for the mother and fetus [17]. However, countries such as Tanzania have minimal policy and health system work addressing prenatal alcohol exposure [18]. Further, the findings regarding the association between alcohol use and perinatal harm are not always communicated to pregnant women [15,18]. The lack of public health interventions aimed at pregnant women and women of childbearing age—including alcohol prevention programs and community-based education on prenatal alcohol exposure—contributes to a significant number of mothers and their fetuses in Tanzania being exposed to alcohol [18]. Nearly 20% of pregnant women in Northern Tanzania and 15% in Dodoma, the capital of Tanzania, report alcohol consumption [18,19]. In Moshi, Tanzania, rates of alcohol use are 2.5 times higher than the national average, which may indicate that similar patterns of alcohol consumption extend to pregnant women [20].

Cultural practices and beliefs are intricately linked to alcohol consumption during pregnancy in SSA, impacting pregnant women's decision to consume alcohol [21]. These cultural traditions and values passed down through generations, constitute knowledge that effectively shapes pregnant individuals' alcohol use practices and can complicate education efforts [18,21,22]. Comprehending the extent and depth of cultural influences on alcohol use during pregnancy is critical to developing culturally appropriate interventions aimed at preventing prenatal alcohol use [14,16,21].

Research investigating the prevalence of prenatal alcohol use in SSA is abundant [6,10,18,19]. Existing research on this topic in Northern Tanzania has assessed female experiences with drinking during pregnancy and the contextual factors contributing to these behaviors [18,19,21]. We aim to expand on this information by exploring the specific consumption patterns and alcohol use practices among pregnant women, the knowledge and attitudes related to prenatal alcohol exposure across different genders and generations, and how community perceptions and cultural beliefs may influence these. Insights from Moshi community members—women and men, young and old—will inform the design of age- and gender-appropriate alcohol reduction interventions and bolster education efforts for reproductive-aged women [23], ultimately aiming to better support pregnant women and reduce alcohol-related pregnancy complications.

Thus, the objectives of this study are to (1) evaluate the proportion of pregnant women who consume alcohol (PWCA), (2) investigate consumption patterns and alcohol use practices of pregnant women and non-pregnant women, accounting for child-bearing age, and (3) assess community knowledge, attitudes, and cultural beliefs on prenatal alcohol use among patients presenting for care to either the Kilimanjaro Christian Medical Centre (KCMC) Emergency Department (ED) or Reproductive Health Center (RHC).

## 2. Methods

### Study design overview

This manuscript represents a sub-study originating from a sequential explanatory mixed-methods study [24] conducted in Moshi, Tanzania—an urban town with over 200,000 residents—focused on exploring gender differences in alcohol use among patients attending the KCMC ED and RHC [25,26].

The high concentration of patients presenting with a chief complaint of injury or physical trauma at KCMC's ED is also a population known for exhibiting high-risk alcohol use

behaviors [27]. Thus, the KCMC ED provides an ideal setting for studying women with frequent or heavy alcohol use [27]. The RHC exclusively serves female patients and offers a representative sample for investigating varying drinking patterns among women [26]. Together, these clinical settings allow for a comprehensive assessment of alcohol use patterns across different demographics of women, particularly pregnant women.

A total of 676 adult patients seeking care at either the KCMC ED or RHC met our inclusion criteria, were approached for participation using a systematic random sampling method and were enrolled. Among these patients, 541 women and 114 men completed the survey. This manuscript's final quantitative dataset comprised 533 women who reported their pregnancy status and age. Out of all 655 participants who completed the survey, 19 were purposively selected—based on trends observed among initial survey responses—to take part in semi-structured in-depth interviews (IDIs).

The larger study aimed to explore gender roles and alcohol use, necessitating an equal number of male and female IDIs. For this sub-study, men were also included to address overarching societal themes and provide a broader, gender-balanced community perspective on prenatal alcohol use. Thus, the IDI participants comprised ten females and nine males. Data collection procedures began in October 2021 and ended in May 2022.

## Quantitative data

**Procedures.** All patients who were (1) over the age of 18 and (2) not prisoners were initially screened for using intake triage registries from the KCMC ED and RHC. Using a systematic random sampling method, every third patient meeting these two criteria was approached. The remaining inclusion criteria—particularly fluency in Kiswahili and the ability to provide informed consent—were assessed upon approaching the patient. The ability to provide consent was defined as being clinically sober, medically stabilized, and physically able to complete the survey verbally.

Female research assistants (RAs) fluent in Kiswahili provided an overview of the study to female patients in a private clinic setting, while male RAs did the same for male patients. This occurred only after patients had achieved medical stability and clinical sobriety. Patients were given the opportunity to inquire about the study, decline participation, or express interest. For those willing to participate, a detailed discussion of the potential risks and benefits took place. Written informed consent was then obtained. Patients who were illiterate marked their initials or a cross mark. Survey questions were read aloud to encourage participation across different literacy levels.

**Instruments.** Quantitative surveys were the primary instruments, assessing baseline sociodemographic factors, self-reported alcohol use, Alcohol Use Disorder Identification Test (AUDIT) scores, Drinker's Inventory of Consequences (DrInC) scores, and Patient Health Questionnaire (PHQ-9) depression scores. Participants were asked to report their alcohol consumption quantity in terms of standard drinks, per WHO guidelines, where one standard drink contains 10 grams of pure ethanol [28].

The Tanzanian research team, fluent in English and Kiswahili and experienced in language translation, translated all basic survey questions on sociodemographic data and self-reported alcohol use from English to Kiswahili. Before data collection, these translations were pilot-tested and revised to ensure grammatical accuracy and the preservation of the original meanings within the basic survey tools.

To comprehensively assess alcohol consumption and unhealthy alcohol-related behaviors among pregnant women, the AUDIT survey tool—with scores ranging from 0 to 40—was employed [29,30]. An AUDIT score of 8 or above indicates clinically significant harmful or hazardous drinking (HHD), in both global and study settings [29,31–34]. Patients with

HHD (AUDIT ≥8) were identified, as their alcohol consumption places them at risk for serious health consequences, necessitating additional alcohol-related treatment and social support [35].

The DrInC survey quantitatively evaluates alcohol-related consequences, with higher scores indicating greater consequences [36]. It includes 50 yes-or-no questions across five domains: physical, intrapersonal, social responsibility, interpersonal, and impulse control [36–38]. The DrInC tool does not measure the severity or frequency of alcohol-related consequences, and no clinically relevant cut-off score exists [38]. This survey was cross-culturally adapted and clinically tested at KCMC before use in this study [38].

The PHQ-9 tool is globally recognized for diagnosing depression, scaled from 0 to 27. It includes nine items assessing depressive symptoms such as appetite changes, sleep patterns, and thoughts of self-harm, with responses to each item scored from 0 to 3 [39–41]. Higher scores indicate more severe depression. Translated into Kiswahili and validated in Tanzania, a score of 9 is the optimal cut-off for identifying major depressive episodes (MDD), classifying patients into depression (PHQ-9 ≥9) and non-depression (PHQ-9 <9) groups [39,40,42–45].

**Analysis.** For this manuscript, the quantitative analysis focused exclusively on survey data from 533 women who reported their pregnancy status and age. Pregnant females were defined as women who had a confirmed in-utero pregnancy based on self-report. Recognizing that some women may still conceive or be at risk for unintended pregnancies at age 45, women of reproductive age were defined as women aged 45 and below [46,47]. This criterion was used as the cut-off value to distinguish between childbearing age (≤45 years; e.g., younger) and non-childbearing age (46+ years; e.g., older) among non-pregnant women.

Descriptive frequencies and statistics were used to compare sociodemographic factors and alcohol use between pregnant women, non-pregnant women of childbearing age (≤45 years), and non-pregnant women not of childbearing age (>45 years). Quantitative measures, including AUDIT, DrInC, and PHQ-9 scores, were also compared between pregnant and non-pregnant women. Continuous data were compared using the Kruskal–Wallis test and Dunn's test, while categorical data were analyzed using Pearson's Chi-square test. Data analyses were performed using user-tested packages in RStudio (version 1.4).

## Qualitative data

**Sampling.** Individuals from the quantitative survey were purposively sampled to participate in semi-structured IDIs, ensuring a diverse range of (1) sociodemographic statuses such as age, marital status, education level, occupation, tribe, and religion, (2) alcohol consumption levels, and (3) perspectives and experiences related to alcohol. For instance, individuals considered for IDIs included those reporting neutral or positive past experiences with alcohol, former alcohol users who are currently abstinent, patients with close friends or family members who consume alcohol heavily, or those who suffered adverse consequences from their own drinking. Our aim was to select a diverse sample of individuals to ensure that the various community perspectives on alcohol use during pregnancy were accurately captured and well-represented in the IDIs.

A gender-balanced approach was used to further ensure a holistic view of community knowledge and personal perspectives on prenatal alcohol use. Specifically, we aimed to interview ten males and ten females, based on qualitative research recommendations suggesting that data saturation is typically reached within a range of 9–17 IDIs for narrowly focused topics [48]. Data saturation, the point at which no new information or themes emerged after three consecutive interviews, was achieved with 19 IDIs, comprising nine males and ten females.

**Procedures.**  Upon completion of the quantitative survey, research assistants invited suitable candidates for IDIs based on predetermined characteristics identified in the survey. Those invited were purposively selected to represent diverse sociodemographic backgrounds, perspectives, and experiences related to alcohol. Consenting participants provided their phone numbers to schedule interviews. To minimize unintentional sampling bias, the study lead reviewed interviewee characteristics monthly, with necessary adjustments in participant selection implemented immediately.

Interviewers were of the same gender as the interviewee and fluent in Kiswahili. Interviewers had previously established rapport with the patients during the survey phase, facilitating open and candid discussions, especially on sensitive and stigmatized topics. Before starting the one-on-one interviews, the study goals were communicated and participants received 5,000 TSH (~2 USD) to cover transportation costs. All interviews were audio-recorded, typically lasting 60–100 min, and included a break with snacks. These semi-structured IDIs were assigned unique identification numbers, ranging from IDI #1 to IDI #19, which were known only to the research team.

**Instrument.**  The interview guide utilized for IDIs included open-ended questions—originally created in English and then translated into Kiswahili—with probes built in for additional questioning when the interviewees' responses were ambiguous, vague, interesting, or inconsistent. The translated, open-ended questions were reviewed and revised for proper phrasing and syntax. These translations were then pilot tested to ensure they were relevant, culturally appropriate in their phrasing, and understandable within the Tanzanian context. The standardized interview guide and related probes were used for all participants to assess community knowledge, views, and perspectives on alcohol use during pregnancy.

**Analysis.**  Qualitative data was analyzed using a classical grounded theory approach [49–52]. Due to the limited research on pregnant female drinking behavior in Moshi, this approach was particularly suitable for capturing and exploring themes arising organically [49,53,54]. The classical grounded theory methodology facilitated the generation of themes directly from the transcripts without relying on preconceived theoretic frameworks, ensuring that the findings were rooted in the participants' experiences and perspectives [49,51,53–55]. The Tanzanian research team was trained on qualitative analysis and interview coding using NVivo 12. An iterative development process using inductive and deductive coding schemes resulted in an initial codebook. To facilitate an ongoing synthesis of data and maintain a dynamic codebook, content memos were created with each emerging theme and subsequent code. To enhance content validity and cultural accuracy, the final analysis was discussed between the entire research team.

## Ethics statement

Prior to data collection, ethical approval for this study was obtained from the Duke University Institutional Review Board, the Kilimanjaro Christian Medical University College Ethical Review Board, and the Tanzanian National Institute of Medical Research. Written consent was formally obtained from all participants. Personal health information used during screening and enrollment was de-identified for confidentiality throughout data collection, storage, analysis, and sharing. Data was only shared in accordance with a data sharing agreement.

## 3.  Results

### Quantitative

The study sample consisted predominantly of women who self-identified as Christian (81.2%) and were affiliated with the Chagga tribe (50.0%), with ages spanning 17–95 years.

On average, pregnant women were 30.1 years of age, with over half of pregnant women between the ages of 25–34 years (Table 1). For women of childbearing age, those who were pregnant were more likely to live with a partner in either a registered (61.2% vs 42.2% in non-pregnant women) or unregistered marriage (29.3% vs 12.9% in non-pregnant women). In comparison, those not pregnant were more likely to be single (39.7% vs 8.6% among pregnant women) or divorced (3.4% vs 0.9% among pregnant women). Younger women (pregnant or non-pregnant) were also more likely to have attained a college-level education (51.7% and 50.8% vs 9.2%), be employed (72.4% and 62.1% vs 47.6%), and have higher personal and household incomes than older, non-childbearing aged women. These values and others can be seen in Table 1.

The majority of pregnant women (56.9%) reported complete abstinence from alcohol, with the remainder drinking at least 1 to 2 times per week (42.2%). Of the pregnant women who drank, 32.8% consumed 1 to 2 standard drinks in a single sitting and 9.5% consumed 3 to 4 standard drinks in a sitting. Non-pregnant women were more likely to drink 3 to 4 standard drinks in a single sitting (16.8% in younger non-pregnant women and 13.5% in older non-pregnant women). Older women not of childbearing age had the highest rates of weekly alcohol use (66.0%), with 17.9% drinking at least 3 to 4 times per week. A higher percentage of older non-pregnant women (55.1%) reported attempts to quit alcohol compared to childbearing-aged non-pregnant women (41.4%) and pregnant women (33.6%). Few younger non-pregnant women (6.5%) and older non-pregnant women (5.4%) sought treatment for their alcohol use. Median [IQR] AUDIT scores were similar across all female groups: 0 [0–2] for pregnant women, 0 [0–4] for younger non-pregnant women, and 0 [0–4] for older non-pregnant women (Table 2).

Younger non-pregnant women reported experiencing significantly more alcohol-related consequences, as indicated by a markedly higher upper quartile in their DrInC scores. Specifically, younger non-pregnant women had a median [IQR] DrInC score of 0 [0–7], compared to the scores of 0 [0–1] for older non-pregnant women and 0 [0–0] for pregnant women (Table 2). Depression severity was highest amongst older non-pregnant women, with a median [IQR] PHQ-9 score of 6 [3–10], followed by 5 [2–8] in younger non-pregnant women and 4 [3–6.25] in pregnant women. Older non-pregnant women (31.4%) were much more likely to be classified as depressed, defined as a PHQ-9 score greater than or equal to 9 points. Similar proportions of all female subgroups report seeking treatment for psychiatric disease, with rates ranging from 5.9% to 12.5% (Table 2).

Most women (70.7%) claimed that consuming 0 alcoholic drinks was the "safe amount" during pregnancy, with 69.4% perceiving prenatal alcohol use as unacceptable throughout all stages of pregnancy. In contrast, over a quarter of older non-pregnant women (25.9%) believed that consuming 1 to 2 standard drinks was a "safe amount" of alcohol during pregnancy (Table 3).

## Qualitative results

Ten women and nine men completed an IDI. Out of the ten women who participated in the IDIs, two had never been pregnant, two were pregnant at the time of the interview, and six had been pregnant in the past. Quotes drawn from these ten women were labeled according to pregnancy status, as either 'Current Pregnancy,' 'Past Pregnancy,' or 'Never Pregnant.' Of the nine men who participated in IDIs, all had either witnessed PWCA in their local communities or had cohabited with their pregnant female partner in a registered or unregistered marriage.

IDI participants ranged in age from 20 to 70 years old and had marital statuses including single or never married, married, divorced, and widowed. Their education statuses ranged

**Table 1.** Study population demographics.

| Demographics stratified by pregnancy status | Overall (n = 533) | Pregnant (n = 116) | Not pregnant, childbearing age (n = 232) | Not pregnant, not childbearing age (n = 185) | P value |
|---|---|---|---|---|---|
| **Age (in years),** *missing: 6 pregnant* | 40.9 (16.0) | 30.1 (5.82) | 31.1 (7.78) | 59.4 (9.98) | **<0.001*** |
| **Age category (in years),** *missing: 6 pregnant* | | | | | **<0.001*** |
| 18–24 | 78 (14.6%) | 19 (16.4%) | 59 (25.4%) | 0 (0%) | |
| 25–34 | 155 (29.1%) | 67 (57.8%) | 88 (37.9%) | 0 (0%) | |
| 35–44 | 101 (18.9%) | 23 (19.8%) | 78 (33.6%) | 0 (0%) | |
| 45–54 | 79 (14.8%) | 1 (0.9%) | 7 (3.0%) | 71 (38.4%) | |
| Over 55 | 114 (21.4%) | 0 (0%) | 0 (0%) | 114 (61.6%) | |
| **Religion,** *missing: 0* | | | | | 0.6886 |
| Christian | 433 (81.2%) | 94 (81.0%) | 186 (80.2%) | 153 (82.7%) | |
| Muslim | 91 (17.1%) | 21 (18.1%) | 43 (18.5%) | 27 (14.6%) | |
| None | 8 (1.5%) | 1 (0.9%) | 3 (1.3%) | 4 (2.2%) | |
| Other | 1 (0.2%) | 0 (0%) | 0 (0%) | 1 (0.5%) | |
| **Marital status,** *missing: 0* | | | | | **<0.001*** |
| Living with a partner in a registered marriage | 263 (49.3%) | 71 (61.2%) | 98 (42.2%) | 94 (50.8%) | |
| Living with a partner, not in a registered marriage | 66 (12.4%) | 34 (29.3%) | 30 (12.9%) | 2 (1.1%) | |
| Never married or single | 169 (31.7%) | 10 (8.6%) | 92 (39.7%) | 67 (36.2%) | |
| Divorced or separate | 23 (4.3%) | 1 (0.9%) | 8 (3.4%) | 14 (7.6%) | |
| Widowed | 12 (2.3%) | 0 (0%) | 4 (1.7%) | 8 (4.3%) | |
| **Highest educational attainment,** *missing: 1* | | | | | **<0.001*** |
| None | 8 (1.5%) | 0 (0%) | 1 (0.4%) | 7 (3.8%) | |
| Primary | 148 (27.8%) | 7 (6.1%) | 33 (14.3%) | 108 (58.4%) | |
| Secondary | 131 (24.6%) | 40 (34.5%) | 58 (25.1%) | 33 (17.9%) | |
| Vocational | 33 (6.2%) | 4 (3.4%) | 14 (6.0%) | 15 (8.1%) | |
| College | 195 (36.6%) | 60 (51.7%) | 118 (50.8%) | 17 (9.2%) | |
| Graduate | 17 (3.2%) | 5 (4.3%) | 8 (3.5%) | 4 (2.2%) | |
| **Employment status,** *missing: 0* | | | | | **<0.001*** |
| Employed | 316 (59.3%) | 84 (72.4%) | 144 (62.1%) | 88 (47.6%) | |
| Unemployed | 156 (29.3%) | 20 (17.2%) | 41 (17.7%) | 95 (51.4%) | |
| Student | 61 (11.4%) | 12 (10.3%) | 47 (20.3%) | 2 (1.1%) | |
| **Tribe,** *missing: 0* | | | | | **<0.01** |
| Chagga | 267 (50.0%) | 64 (55.2%) | 97 (41.8%) | 106 (57.3%) | |
| Pare | 50 (9.4%) | 9 (7.8%) | 16 (6.9%) | 25 (13.5%) | |
| Sukuma | 26 (4.9%) | 8 (6.9%) | 13 (5.6%) | 5 (2.7%) | |
| Massai | 21 (3.9%) | 5 (4.3%) | 8 (3.4%) | 8 (4.3%) | |
| Sambaa | 16 (3.0%) | 3 (2.6%) | 9 (3.9%) | 4 (2.2%) | |
| Nyaturu | 17 (3.2%) | 3 (2.6%) | 13 (5.6%) | 1 (0.5%) | |
| Other African | 135 (25.3%) | 24 (20.7%) | 76 (32.8%) | 35 (18.9%) | |
| Non-African | 1 (0.2%) | 0 (0%) | 0 (0%) | 1 (0.5%) | |
| **Personal income (in TZS),** *missing: 3* | | | | | **<0.001*** |
| 0 to 50000 | 176 (33.0%) | 20 (17.2%) | 66 (28.4%) | 90 (48.6%) | |
| 50001 to 100000 | 70 (13.1%) | 14 (12.1%) | 32 (13.8%) | 24 (13.0%) | |
| 100001 to 150000 | 32 (6.0%) | 7 (6.0%) | 14 (6.0%) | 11 (5.9%) | |
| 150001 to 200000 | 45 (8.4%) | 13 (11.2%) | 22 (9.5%) | 10 (5.4%) | |
| >200000 | 207 (38.8%) | 61 (52.6%) | 97 (41.8%) | 49 (26.5%) | |

*(Continued)*

**Table 1.** (Continued)

| Demographics stratified by pregnancy status | Overall (n = 533) | Pregnant (n = 116) | Not pregnant, childbearing age (n = 232) | Not pregnant, not childbearing age (n = 185) | P value |
|---|---|---|---|---|---|
| **Household income, *missing: 3*** | | | | | **<0.010**** |
| 0 to 50000 | 47 (8.8%) | 3 (2.6%) | 14 (6.0%) | 30 (16.2%) | |
| 50001 to 100000 | 56 (10.5%) | 8 (6.9%) | 18 (7.8%) | 30 (16.2%) | |
| 100001 to 150000 | 31 (5.8%) | 6 (5.2%) | 5 (2.2%) | 20 (10.8%) | |
| 150001 to 200000 | 63 (11.8%) | 9 (7.8%) | 35 (15.1%) | 19 (10.3%) | |
| >200000 | 333 (62.5%) | 89 (76.7%) | 159 (68.5%) | 85 (45.9%) | |
| **Number of people in household, *missing: 1*** | 3.62 (1.79) | 3.55 (1.55) | 3.57 (1.93) | 3.71 (1.75) | 0.655 |

Unadjusted associations by pregnancy status and age. data are shown as [1]N (%) for categorical variables and mean (SD) for continuous variables. Continuous data were compared using analysis of variance (ANOVA) type 3 sum of squares. Categorical data were compared using Pearson's Chi-square test. The symbol (***) denotes that a P value is statistically significant at the $\alpha$ = 0.001 level. The symbol (**) denotes that a P value is statistically significant at the $\alpha$ = 0.01 level. The symbol (*) denotes that a P value is statistically significant at the $\alpha$ = 0.05 level. For the variable Tribe, the sub-category of other African primarily includes the following tribes: Iraq, Mmeru, Muha, among others not mentioned.

**Table 2. Alcohol use practices, consequences and depression status by pregnancy status.**

| Alcohol use practices by pregnancy status | Overall (n = 533) | Pregnant (n = 116) | Not pregnant, child-bearing age (n = 232) | Not pregnant, not child-bearing age (n = 185) | P value |
|---|---|---|---|---|---|
| **Drinking frequency (n, %), *missing: 2*** | | | | | **<0.001**** |
| 0 times/week | 243 (45.6%) | 66 (56.9%) | 115 (49.6%) | 62 (33.5%) | |
| 1 to 2 times/week | 227 (42.6%) | 45 (38.8%) | 93 (40.1%) | 89 (48.1%) | |
| 3 to 4 times/week | 43 (8.1%) | 4 (3.4%) | 20 (8.6%) | 19 (10.3%) | |
| 5 to 6 times/week | 4 (0.8%) | 0 (0%) | 2 (0.9%) | 2 (1.1%) | |
| Everyday | 14 (2.6%) | 0 (0%) | 2 (0.9%) | 12 (6.5%) | |
| **Drinking quantity (# standard drinks per sitting) (n, %) *missing: 1*** | | | | | **<0.001**** |
| 0 standard drinks | 244 (45.8%) | 66 (56.9%) | 115 (49.6%) | 63 (34.1%) | |
| 1 to 2 standard drinks | 202 (37.9%) | 38 (32.8%) | 72 (31.0%) | 92 (49.7%) | |
| 3 to 4 standard drinks | 75 (14.1%) | 11 (9.5%) | 39 (16.8%) | 25 (13.5%) | |
| 5 to 6 standard drinks | 10 (1.9%) | 0 (0%) | 5 (2.2%) | 5 (2.7%) | |
| >6 standard drinks | 1 (0.2%) | 0 (0%) | 1 (0.4%) | 0 (0%) | |
| **Weekly alcohol expenses (TZS) (n, %), *missing: 1*** | | | | | **<0.01**** |
| 0 to 10000 | 460 (86.3%) | 98 (84.5%) | 190 (81.9%) | 172 (93.0%) | |
| 10001 to 50000 | 69 (12.9%) | 17 (14.7%) | 39 (16.8%) | 13 (7.0%) | |
| 50001 to 100000 | 3 (0.6%) | 0 (0%) | 3 (1.3%) | 0 (0%) | |
| **Alcohol preferences (n, %), *missing: 1*** | | | | | **<0.001**** |
| Wine | 67 (12.6%) | 24 (20.7%) | 37 (15.9%) | 6 (3.2%) | |
| Light Beer | 38 (7.1%) | 8 (6.9%) | 24 (10.3%) | 6 (3.2%) | |
| Beer | 106 (19.9%) | 15 (12.9%) | 39 (16.8%) | 52 (28.1%) | |
| Mbege (banana-based beer) | 67 (12.6%) | 2 (1.7%) | 13 (5.6%) | 52 (28.1%) | |
| Other | 12 (2.4%) | 0 (0%) | 1 (0.4%) | 6 (3.2%) | |
| None | 240 (45.0%) | 66 (56.9%) | 112 (48.3%) | 62 (33.5%) | |
| **Attempted quitting at any time-point (n, %), *missing: 4*** | | | | | **<0.01**** |
| No | 292 (54.8%) | 75 (64.7%) | 135 (58.2%) | 82 (44.3%) | |
| Yes | 237 (44.5%) | 39 (33.6%) | 96 (41.4%) | 102 (55.1%) | |
| **Sought treatment for alcohol use (n, %), *missing: 4*** | | | | | 0.063 |
| Yes | 25 (4.7%) | 0 (0%) | 15 (6.5%) | 10 (5.4%) | |
| No | 504 (94.6%) | 116 (100%) | 215 (92.7%) | 173 (93.5%) | |

*(Continued)*

**Table 2.** (Continued)

| Alcohol use practices by pregnancy status | Overall (n = 533) | Pregnant (n = 116) | Not pregnant, child-bearing age (n = 232) | Not pregnant, not child-bearing age (n = 185) | P value |
|---|---|---|---|---|---|
| **AUDIT score (median, IQR), *missing: 0*** | 0 (0–4) | 0 (0–2) | 0 (0–4) | 0 (0–4) | **0.113** |
| **HHD status (n, %), *missing: 0*** | | | | | 0.092 |
| Yes | 69 (12.9%) | 10 (8.6%) | 38 (16.4%) | 21 (11.4%) | |
| No | 464 (87.1%) | 106 (91.4%) | 194 (83.6%) | 164 (88.6%) | |
| **DrInC score (median, IQR), *missing: 0*** | 0 (0–3) | 0 (0–0) | 0 (0–7) | 0 (0–1) | **<0.01\*\*** |
| **PHQ9 score (median, IQR), *missing: 0*** | 5 (3–8) | 4 (3–6.25) | 5 (2–8) | 6 (3–10) | **<0.001\*\*\*** |
| **Depression (PHQ-9 ≥9) (n, %), *missing: 0*** | | | | | **<0.001\*\*\*** |
| Yes | 127 (23.8%) | 14 (12.1%) | 55 (23.7%) | 58 (31.4%) | |
| No | 406 (76.2%) | 102 (87.9%) | 177 (76.3%) | 127 (68.6%) | |
| **Sought psychiatric treatment (n, %), *missing: 3*** | | | | | 0.094 |
| Yes | 51 (9.4%) | 11 (9.5%) | 29 (12.5%) | 11 (5.9%) | |
| No | 487 (90.0%) | 105 (90.5%) | 200 (86.2%) | 174 (94.1%) | |

Unadjusted associations with alcohol use practices by pregnancy status and age. Data are shown as N (%) for categorical variables and median (IQR) for continuous variables. Continuous data were compared using Kruskal–Wallis test followed by Dunn's test. Categorical data were compared using Pearson's Chi-square test. The symbol (\*\*\*) denotes that a P value is statistically significant at the α = 0.001 level. The symbol (\*\*) denotes that a P value is statistically significant at the α = 0.01 level. The symbol (\*) denotes that a P value is statistically significant at the α = 0.05 level. For the variable alcohol preferences, the sub-category of other primarily includes the following: Liquor/Spirits, Daddi, Ulanzi, Piwa, Gongo, Changaa, among others not mentioned.

**Table 3. Survey-reported knowledge and attitudes on alcohol use by pregnancy status.**

| Knowledge and attitudes on alcohol use by pregnancy status | Overall (n = 533) | Pregnant (n = 116) | Not pregnant, child-bearing age (n = 232) | Not pregnant, not child-bearing age (n = 185) | P value |
|---|---|---|---|---|---|
| **Safe amount of alcohol while pregnant, *missing: 81*** | | | | | **<0.001\*\*\*** |
| 0 standard drinks | 377 (70.7%) | 83 (71.6%) | 175 (75.4%) | 119 (64.3%) | |
| 1 to 2 standard drinks | 70 (13.1%) | 6 (5.2%) | 16 (6.9%) | 48 (25.9%) | |
| 3 to 4 standard drinks | 3 (0.6%) | 0 (0%) | 1 (0.4%) | 2 (1.1%) | |
| ≥10 standard drinks | 2 (0.4%) | 1 (0.9%) | 0 (0%) | 1 (0.5%) | |
| **When alcohol use is acceptable during pregnancy, *missing: 85*** | | | | | **<0.001\*\*\*** |
| It is never okay | 370 (69.4%) | 85 (73.3%) | 171 (73.7%) | 114 (61.6%) | |
| Beginning of pregnancy | 17 (3.2%) | 1 (0.9%) | 7 (3.0%) | 9 (4.9%) | |
| Middle of pregnancy | 7 (1.3%) | 0 (0%) | 3 (1.3%) | 4 (2.2%) | |
| End of pregnancy | 16 (3.0%) | 1 (0.9%) | 5 (2.2%) | 10 (5.4%) | |
| It is okay at any time | 38 (7.1%) | 2 (1.7%) | 6 (2.6%) | 30 (16.2%) | |
| **Perceived alcohol use as unhealthy, *missing: 8*** | | | | | 0.068 |
| Yes | 363 (68.1%) | 67 (57.8%) | 166 (71.6%) | 130 (70.3%) | |
| No | 162 (30.4%) | 47 (40.5%) | 62 (26.7%) | 53 (28.6%) | |

Unadjusted associations with knowledge and attitudes on alcohol use by pregnancy status and age. Data are shown as [1]N (%) for categorical variables and mean (SD) for continuous variables. Continuous data were compared using analysis of variance (ANOVA) test with type 3 sum of squares. Categorical data were compared using Pearson's Chi-square test. The symbol (\*\*\*) denotes that a P value is statistically significant at the α = 0.001 level. The symbol (\*\*) denotes that a P value is statistically significant at the α = 0.01 level. The symbol (\*) denotes that a P value is statistically significant at the α = 0.05 level.

from primary to college level. Alcohol consumption patterns ranged from complete abstinence to frequent consumption, such as 3 to 4 bottles per sitting nearly 5 to 6 times per week. Three themes were deductively and inductively identified from the 19 interviews: (1) Observation of prenatal alcohol use in Moshi, including its commonality, (2) Knowledge of the consequences related to alcohol use during pregnancy, such as the influence of cultural beliefs and perceived harms, and (3) Community attitudes regarding PWCA (Table 4).

**Table 4. Qualitative themes and sub-themes regarding alcohol use and pregnancy.**

| Themes | Sub-themes |
| --- | --- |
| Observations of prenatal alcohol use in Moshi | Commonality of prenatal alcohol use in Moshi due to widespread disbelief of associated harms |
| Knowledge of the consequences related to alcohol use during pregnancy | Perceived cultural benefits of alcohol use during pregnancy and delivery |
| | Perceived harms of prenatal alcohol exposure |
| Community attitudes regarding PWCA | Negative community views of prenatal alcohol use |
| | Rationalizations for why pregnant women consume alcohol |

## Observations of prenatal alcohol use in Moshi

**Commonality of prenatal alcohol use in Moshi due to widespread disbelief of associated harms.** Over half of the IDI participants noted that it was "*quite common to see a pregnant woman drinking alcohol*" (IDI #6, Female, Past Pregnancy) in Moshi, with almost half of the female interviewees reporting alcohol consumption themselves during a current, or past pregnancy. One participant recalled:

> "*When I was pregnant, I used to go to the clinic and being told not to drink alcohol but when I returned home, I continued to drink alcohol as usual.*" (IDI #6, Female, Past Pregnancy)

Another female respondent shared her observations of PWCA, which shaped her own alcohol use.

> "*I have seen many … [pregnant women] who use alcohol, some of them even drink a lot of beer bottles but I have never witnessed any of them getting any bad effect even after delivery …*
>
> *Drinking wine helps me with my health, such as in food digestion process. That's why I decided to continue drinking even when pregnant … I will drink just 2 glasses of wine per week. It's such a little amount that am sure cannot bring harm to my baby.*" (IDI #19, Female, Current Pregnancy)

Several other participants witnessed a lack of observed adverse consequences when consuming alcohol while pregnant, which appeared to contribute to the commonality of this practice in Moshi. This sentiment is accurately summarized by one participant who highlights that a mother gave birth to a healthy baby after consuming alcohol throughout her pregnancy.

> "*There was a woman; when she was pregnant, she drank so much alcohol. One day, the baby turned up, we brought her here to KCMC hospital. She was drunk and she was told it was still not the time to give birth. She returned home and continued drinking alcohol until her delivery, yet she still gave birth to a healthy baby. Right now, the child is three years old.*" (IDI #6, Female, Past Pregnancy)

Another participant had a similar sentiment, which describes how a mother gave birth to an unhealthy baby while not drinking during pregnancy and a healthy baby while drinking during pregnancy.

> "*During the first pregnancy, she did everything … she had to do, she even stopped drinking alcohol, but the baby was still born with problems. After giving birth to the first child, she started to drink alcohol, even when she was pregnant with the second child… She refused to*

*stop drinking alcohol until she gave birth…and she had a normal child with no problems.*"
(IDI #9, Female, Past Pregnancy)

These results highlight the commonality of alcohol use during pregnancy in Moshi, Tanzania, largely due to the unobserved harms associated with this practice among community members. In contrast to these excerpts, the remaining IDI participants stated that alcohol use during pregnancy was "*not a common practice*" (IDI #11, Male) in Moshi due to community sensitization to alcohol-related harms on the mother and fetus, whereby "*people are aware of the effects of alcohol during pregnancy*" (IDI #11, Male) and "*they understand the effects of alcohol*" (IDI #5, Male).

## Knowledge of consequences related to alcohol use during pregnancy

**Perceived cultural benefits of alcohol use during pregnancy and delivery.** Cultural beliefs surrounding prenatal alcohol use had both tribal and cultural origins. For example, nearly a quarter of participants highlighted that Chagga tribe members encouraged moderate alcohol consumption before, during, and after delivery. These norms encouraged pregnant women to "*sip a small amount of alcohol*" (IDI #8, Male), as this was thought to provide benefits to the pregnant female and unborn child. Specifically, it was thought that alcohol would "*help the baby [to] grow big and become active*" (IDI #11, Male), with participants encouraging alcohol use "*at the middle of pregnancy*" because "*at that stage, the baby should grow and be strong before delivery*" (IDI #8, Male).

Participants likewise highlighted numerous perceived benefits of alcohol use during labor, such as providing the mother with "*increased power*" (IDI #7, Female, Past Pregnancy) to push the baby through the birth canal, alleviating stress, "*[reducing] labor pain*" (IDI #14, Male) and "*abdominal cramps*" (IDI #12, Male), and "*increase milk production for [the] baby to breast-feed*" (IDI #7, Female, Past Pregnancy). One female interviewee elaborated on the perceived benefit of alcohol for breast milk production, recounting that:

"*There is a notion where I live that taking alcohol during and after delivery increases milk production and eventually the baby gets enough milk to breastfeed… after my third delivery where my grandmother insisted for me to take it in order to increase milk production for my baby to breastfeed.*" (IDI #7, Female, Past Pregnancy)

These beliefs, however, were not characteristic of all parts of Moshi. One participant highlights a distinct discrepancy between village views and town views concerning PWCA, explaining that alcohol use during pregnancy is prevalent and accepted only in certain regional contexts:

"*The community views [PWCA] differently. If you ask people from the village, they would say its right, but in town, because people are aware of the effect of alcohol, you may find few of them who drink…*" (IDI #14, Male)

These results highlight the impact of cultural beliefs on the alcohol consumption patterns of pregnant women and perceived benefits.

**Perceived harms of prenatal alcohol exposure.** While cultural beliefs may influence the use of alcohol during pregnancy, many participants noted substantial harms of prenatal alcohol exposure on the developing fetus, including premature birth, fetal anomalies, blindness, nutritional deficiencies, and neurological, cognitive, and motor disabilities. One respondent notes:

"*The baby may be born with some abnormalities such as blindness, mental problems or born prematurely. I have seen one female who was drinking during pregnancy, and she end up delivering a premature baby.*" (IDI #11, Male)

A handful of participants suggested that alcohol-induced appetite loss in expectant mothers could lead to nutritional deficiencies hindering proper fetal development.

"*Alcohol sometimes causes someone to lose appetite, it depends on the type of alcohol you drink. You may drink alcohol and fail to eat properly, and harm the baby or the mother can get anemia.*" (IDI #9, Female, Past Pregnancy)

The impact of alcohol use was seen to extend into the postnatal period. About a quarter of interviewees cautioned against alcohol use during breastfeeding, noting that "*it will have [an] impact on [the] child`s mental and motor development and eventually restricts normal and proper growth of that child*" (IDI #7, Female, Past Pregnancy).

These risks extended to the mother as well, with a fifth of interviewees asserting that alcohol use during pregnancy can have a negative impact on female health, such as engaging in unsafe sexual behaviors or disrupting motor function, where an intoxicated pregnant women may "*fall down and injure the stomach… [with] a possibility of terminating [the] pregnancy*" (IDI #14, Male). One interviewee explained:

"*In general, alcohol consumption during pregnancy is not good for a pregnant woman to drink too much alcohol. When got drunkard she can do unprotected sex with someone she does not know and put herself at risk of sexual transmitted infections or AIDS during pregnancy.*" (IDI #9, Female, Past Pregnancy)

Finally, a few interviewees emphasized the importance of adequate education on the detrimental effects of prenatal alcohol use, noting that "*[Women] are always ready to try their best when they are given right knowledge and education about the consequences of alcohol to themselves until delivery time and to their unborn babies*" (IDI #16, Female, Never Pregnant).

## Community attitudes regarding PWCA

**Negative community views of prenatal alcohol use.** Generally, community perspectives cast PWCA in a negative light. Most IDI participants claimed that alcohol consumption during pregnancy was prohibited and engaging in such behaviors would quickly attract social disapproval. To avoid censure, one respondent specifically mentioned that PWCA tended to "*drink privately*" (IDI #8, Male). Interviewees described that PWCA are viewed as incapable of suppressing alcoholic urges and of making sacrifices for the sake of her child. This shared community attitude was epitomized by the language used by respondents to describe PWCA who "*does not wish good for her baby*" (IDI #3, Female, Never Pregnant). Participants claimed that the community views PWCA as "*foolish*" (IDI #10, Female, Never Pregnant) and "*taboo*" (IDI #9, Female, Past Pregnancy) whereby alcohol consumption while pregnant is "*careless*" (IDI #3, Female, Never Pregnant). Additionally, one participant stated, "*The community perceive them as 'risk taker', 'don't care' they can even be isolated in the community*" (IDI #4, Male), thereby demonstrating the profound influence of these adverse community perceptions on PWCA.

Several participants compared PWCA to non-pregnant women who consume alcohol, implying that PWCA are viewed particularly poorly. Specifically, participants posited that since pregnant women actively "*agreed to conceive*" (IDI #9, Female, Past Pregnancy), they

and their partner were responsible for ensuring the well-being and health of themselves and of the unborn child. For example, participants remarked that "*whether she is educated or not education, a pregnant woman drinking alcohol will seem it is it so wrong*" (IDI #9, Female, Past Pregnancy) and when a women drinks while pregnant, "*[the community] normally does some traditional ritual to her*" (IDI #11, Male).

**Rationalizations for why pregnant women consume alcohol.** Several factors, collectively, were rationalized by participants to cause a pregnant woman to either increase, decrease, or altogether abstain from alcohol consumption during pregnancy. First, nearly half of the interviewees associated prenatal alcohol use with an inability to stop consumption due to pre-existing alcohol dependency. Interviewees hypothesized that stress occurring before a woman became pregnant underlied pre-pregnancy alcohol use.

> "*Some of them were good drinkers even before pregnancy therefore it's not easy for them to stop abruptly during pregnancy… I think it's just because she was already used to taking alcohol regularly when she wasn't pregnant, and she failed to stop using.*" (IDI #3, Female, Never Pregnant)

> "*For sure I have no idea of why she takes alcohol at that [perinatal] period, but I know she has been using all that since long time ago, at some point she had family conflicts with her husband which may be one of the reasons.*" (IDI #7, Female, Past Pregnancy)

Moreover, participants held the belief that pregnant women are likely to increase alcohol consumption if they experienced an unexpected pregnancy while unmarried. Such situations were thought to induce stress and societal pressures for these expectant mothers, as elucidated by one participant's remark:

> "*We have one problem in our community, most men run away once they have [kids]. So due to stress, this women start or increase the amount of drinking alcohol.*" (IDI #18, Male)

Second, over a quarter of participants believed that pregnancy hormones may cause a "*strong desire and cravings*" (IDI #3, Female, Never Pregnant) for certain foods and drinks, including alcohol. For example, one participant explained how pregnancy cravings may influence alcohol use:

> "*There are quite a number of pregnant women who take alcohol. Depending on their desires and pregnancy cravings, some drink a lot, others due to pregnancy don't even want to smell [alcohol], others remain using as they were before.*" (IDI #7, Female, Past Pregnancy)

## 4. Discussion

This study aimed to explore perceptions and assess current rates of alcohol use among pregnant women in Moshi. To the best of our knowledge, this is one of the first studies using a mixed-methods approach to closely examine community attitudes around alcohol use during pregnancy in Moshi, Tanzania. Our analysis found that rates of alcohol use among pregnant women in Moshi remain high, with the biggest predictor of prenatal alcohol use being pre-pregnancy alcohol use. While overall, the community held negative views around PWCA and believed women should not drink at any point in their pregnancy, some socio-cultural beliefs encouraged alcohol consumption during pregnancy. The high rates of alcohol use and complex factors influencing use point to the need for robust community-wide health

messaging coupled with patient-level, pre-pregnancy education and holistic support in order to reduce this generational harm.

Prenatal alcohol use in Moshi remains notably elevated. We found that 42% of pregnant women attending KCMC's ED and RHC consume alcohol weekly, drinking 1 to 2 standard drinks of alcohol per sitting, with previous alcohol use being most associated with use during pregnancy. Our qualitative results supported the commonality of alcohol use during pregnancy, with this behavior noted to be frequently observed in local communities, spurred by a lack of observed consequences. These rates are significantly higher than previous, nearby studies that found alcohol consumption during pregnancy to be 20% in Northern Tanzania and 15% in neighboring Dodoma [18,19]. These findings, particularly the ubiquitousness of prenatal alcohol use, are cause for concern given existing recommendations that pregnant women should abstain from alcohol entirely [17]. Populations exhibiting high rates of alcohol consumption are typically associated with a high prevalence of FASD; however, within Tanzania, like most of SSA, the true prevalence of FASD is not well understood and corresponding literature is rare [8–11]. FASD and other alcohol-related congenital anomalies typically develop early in pregnancy, often before women realize they are pregnant [56,57]. Importantly, alcohol use behaviors before pregnancy were identified as a rationalization for alcohol use during pregnancy. Several participants in the IDIs noted that pre-pregnancy alcohol habits were the primary reason for continued alcohol use in pregnancy. This is especially important to consider as young women who are currently not, but may become pregnant, exhibited substantial drinking frequencies (50.5% consuming at least once per week), drinking quantities (50.4% consuming at least one standard drink per sitting), and alcohol-related consequences (DrInC = 0 [0–7]). Previous studies have identified pre-pregnancy alcohol use among young women to be a significant predictor of alcohol use during pregnancy, with literature noting that the minimization of these habits are key in reducing prenatal alcohol exposure [19,58].

In both our quantitative and qualitative analysis, prenatal alcohol use was generally censured, however, sociocultural and community observations and beliefs underlied continued alcohol use during pregnancy. Survey data revealed that 70% of participants found any amount of alcohol to be unsafe at any point during pregnancy, a finding qualitatively supported by the general negative community views of PWCA and perceived harms of this behavior. The observed societal stigma towards women who drink, regardless of pregnancy, warrants further investigation to determine if this stigma is more profound in pregnancy, or applies generally to all women. Aside from previous alcohol use, IDI responses provided a more nuanced understanding of why prenatal alcohol use continued within Moshi, mentioning stress, addiction, and observations of other women who drank and gave birth to healthy children. Accessing mental health support is challenging in Moshi, Tanzania, particularly for women [59–62]. In our quantitative sample of women, only a small percentage (9.4%) sought psychiatric treatment programs, despite a high prevalence of depression (23.8%). Previous literature has also found stress and depression to be highly prevalent in women from SSA both during and after pregnancy [61,63–65], with stress also linked to increased alcohol consumption both in this region and worldwide [24,62,66–69]. The influence of depression, stress, and addiction underlying alcohol consumption during pregnancy points to a need for future programming to integrate accessible mental health care components into reproductive health services, and the minimization of alcohol use among both pregnant and non-pregnant women of childbearing age [62,67,70–72].

These factors illustrate the necessity of increased community-wide health messaging centered around the dangers of prenatal alcohol exposure, coupled with patient-level,

pre-pregnancy education and holistic support for women of childbearing age. For example, our analysis found that persistent beliefs exist, painting prenatal alcohol use as beneficial, which illustrates that a gap remains in community education and practice. This call to action joins other literature, which similarly has identified high rates of alcohol use during pregnancy in SSA, to put greater focus on developing and implementing behavioral interventions [6,19,24,58,73]. Interventions to reduce alcohol consumption during pregnancy have been tried and tested worldwide, with formats being primarily medical-, psychosocial-, and education-centered [74]. The most effective interventions for pregnant women were psychosocial-based, specifically brief interventions (BIs), which is a focused, motivational interview delivered directly to the patient and can be readily incorporated into prenatal care visits [75]. This intervention type may be especially useful in Moshi, as BIs have been shown to be particularly effective in resource-constrained settings, requiring little cost to operate and can be performed by lay healthcare workers [76]. Future programming in this region should incorporate both increased community messaging alongside effective education and support for expectant mothers.

Local policy efforts should be aware of the high rates of alcohol use still present in Moshi and re-focus health priorities on mitigating this preventable harm. Additionally, to better inform the creation of appropriate interventions and programming, future studies should investigate the extent of current alcohol-related fetal harm in Tanzania to ensure the most impactful prevention strategies are utilized. Through these strategies, alcohol-related harm can be mitigated, with improved health outcomes for present and future generations.

### Strengths and limitations

The primary limitations of this study include selection bias, as the population was recruited solely from one health facility in Moshi which may not adequately represent the broader Moshi community. Another limitation includes recall bias, as alcohol use behaviors were self-reported through surveys. That being said, this one hospital evaluation will serve as a contextual framework for an implementation study and further literature about barriers to alcohol harm reduction in low resource settings. Additionally, it is important to acknowledge the uncertainty surrounding the observed prevalence of pregnant women who consume alcohol (42.2%) given the limited size of our sample of pregnant women (n = 116).

## 5. Conclusion

Our study highlights both current high rates of alcohol consumption among pregnant women in Moshi, a behavior shaped in part by cultural beliefs and practices, alongside negative community attitudes towards these individuals. Stress coping, existing alcohol dependency, and pregnancy-related cravings constitute additional motivations for alcohol consumption among pregnant women. These findings stress the need for future supportive, education-based programming for young women of childbearing age. More holistic pregnancy-related healthcare services and infrastructure in this region may be the solution to curbing alcohol use and improving health outcomes within the Moshi community.

## Supporting information

**S1 Checklist. STROBE statement.**
(DOCX)

**S2 Checklist. Inclusivity in global research.**
(DOCX)

## Author contributions

**Conceptualization:** Kirstin West, Alena Pauley, Sharla Rent, Catherine A. Staton.

**Data curation:** Alena Pauley, Miriana Mikindo, Yvonne Sawe, Joseph Kilasara.

**Formal analysis:** Kirstin West.

**Funding acquisition:** Alena Pauley, Catherine A. Staton.

**Investigation:** Kirstin West, Alena Pauley, Francis Sakita, Sharla Rent, Bariki Mchome, Blandina T. Mmbaga, Catherine A. Staton.

**Methodology:** Kirstin West, Alena Pauley, Sharla Rent, Catherine A. Staton.

**Supervision:** Sharla Rent, Blandina T. Mmbaga, Catherine A. Staton.

**Visualization:** Kirstin West.

**Writing – original draft:** Kirstin West, Alena Pauley, Mia Buono, Sharla Rent.

**Writing – review & editing:** Kirstin West, Alena Pauley, Francis Sakita, Sharla Rent, Bariki Mchome, Blandina T. Mmbaga, Catherine A. Staton.

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
