## [Decision Letter · Decision Letter 0]

3 Sep 2024

PGPH-D-24-01850

The Burden of Generational Harm due to Alcohol use in Tanzania: a mixed method study of pregnant women

Dear Dr. Staton,

Thank you for submitting your manuscript to PLOS Global Public Health. After careful consideration, we feel that it has merit but does not fully meet PLOS Global Public Health’s publication criteria as it currently stands. Therefore, we invite you to submit a revised version of the manuscript that addresses the points raised during the review process.

We look forward to receiving your revised manuscript.

Kind regards,

Hani Mowafi, M.D., M.P.H.

Academic Editor

Journal Requirements:

Additional Editor Comments (if provided):

Thank you for the submission. There were several comments from reviewers that need to be addressed prior before the article can be published at PLOS Global Health. Please review them in detail and provide responses at your earliest convenience.

Reviewers' comments:

Reviewer's Responses to Questions

**Comments to the Author**

1. Does this manuscript meet PLOS Global Public Health’s publication criteria ? Is the manuscript technically sound, and do the data support the conclusions? The manuscript must describe methodologically and ethically rigorous research with conclusions that are appropriately drawn based on the data presented.

Reviewer #1: Yes

Reviewer #2: Partly

2. Has the statistical analysis been performed appropriately and rigorously?

Reviewer #1: Yes

Reviewer #2: I don't know

3. Have the authors made all data underlying the findings in their manuscript fully available (please refer to the Data Availability Statement at the start of the manuscript PDF file)?

Reviewer #1: Yes

Reviewer #2: Yes

4. Is the manuscript presented in an intelligible fashion and written in standard English?

Reviewer #1: Yes

Reviewer #2: No

5. Review Comments to the Author

Reviewer #1: I congratulate the authors about this interesting paper.

I have only as recommendation to include the entire significance of abbreviations across the paper to facilitate the reading.

Also I recommended the authors to include the explanation of KCMC ED and RHC in abbreviations

Reviewer #2: Thank you for letting me review your article, it was an interesting read, and it is a topic worth writing about. I have a large amount of comments, see below. One of the main things I would suggest is to proofread your article again, as there are a multitude of grammatical errors. Furthermore, it is unclear how the research has been conducted. Multiple researches appear to have been conducted simultaneously, and then the data has been sifted through to create this article. The sentence “A thorough overview of the methods has been previously published 17.” made this research all the more confusing. There were also multiple instances of either missing references or incorrect references. I was unable to finish my review due to the large number of issues I found thus far. I have yet to review the tables and the qualitative results.

6. PLOS authors have the option to publish the peer review history of their article (what does this mean? ). If published, this will include your full peer review and any attached files.

**Do you want your identity to be public for this peer review?** For information about this choice, including consent withdrawal, please see our Privacy Policy .

Reviewer #1: No

Reviewer #2: **Yes: ** dr. Peter Johan Kruithof

---

## [Decision Letter · Decision Letter 1]

20 Nov 2024

The Burden of Generational Harm due to Alcohol use in Tanzania: a mixed method study of pregnant women

PGPH-D-24-01850R1

Dear Dr. Staton,

We are pleased to inform you that your manuscript 'The Burden of Generational Harm due to Alcohol use in Tanzania: a mixed method study of pregnant women' has been provisionally accepted for publication in PLOS Global Public Health.

Best regards,

Hani Mowafi, M.D., M.P.H.

Academic Editor

Thank you for addressing the revisions noted by the previous review. Congratulations on the article's acceptance for publication.

Reviewer Comments (if any, and for reference):

Reviewer's Responses to Questions

**Comments to the Author**

1. If the authors have adequately addressed your comments raised in a previous round of review and you feel that this manuscript is now acceptable for publication, you may indicate that here to bypass the “Comments to the Author” section, enter your conflict of interest statement in the “Confidential to Editor” section, and submit your "Accept" recommendation.

Reviewer #2: All comments have been addressed

2. Does this manuscript meet PLOS Global Public Health’s publication criteria ? Is the manuscript technically sound, and do the data support the conclusions? The manuscript must describe methodologically and ethically rigorous research with conclusions that are appropriately drawn based on the data presented.

Reviewer #2: Yes

3. Has the statistical analysis been performed appropriately and rigorously?

Reviewer #2: Yes

4. Have the authors made all data underlying the findings in their manuscript fully available (please refer to the Data Availability Statement at the start of the manuscript PDF file)?

Reviewer #2: No

5. Is the manuscript presented in an intelligible fashion and written in standard English?

Reviewer #2: Yes

6. Review Comments to the Author

Reviewer #2: Dear authors,

Let me start by thanking you for taking the effort to address the comments I had made. I understand that this would have taken significant time and effort. The article has been greatly improved.

May also apologise for the tone of my previous review.

Kind regards,

Dr. Peter J. Kruithof

7. PLOS authors have the option to publish the peer review history of their article (what does this mean? ). If published, this will include your full peer review and any attached files.

**Do you want your identity to be public for this peer review?** For information about this choice, including consent withdrawal, please see our Privacy Policy .

Reviewer #2: **Yes: ** Dr. Peter J. Kruithof
